# Seasonal Variations in the Starch Properties of Sweet Potato Cultivars

**Thaís Paes Rodrigues dos Santos** [1] , **Magali Leonel** [1,*] , **Luciana Alves de Oliveira** [2] ,
**Adalton Mazetti Fernandes** [1] , **Sarita Leonel** [1,3] **and Jason Geter da Silva Nunes** [1]

1   Center for Tropical Roots and Starches (CERAT), São Paulo State University (UNESP),
    Botucatu 18610307, Brazil
2   Embrapa Cassava and Fruits, Cruz das Almas 44380000, Brazil
3   School of Agriculture (FCA), São Paulo State University (UNESP), Botucatu 18610307, Brazil
*   Correspondence: magali.leonel@unesp.br; Tel.: +55-14-3880-7612

**Abstract:** Starch is widely used in the food and non-food industries, and this is related to its physico-chemical characteristics. In the coming years, climate changes will become unpredictable, and these conditions may affect the process of starch biosynthesis and polymer properties. The sweet potato starch market has grown substantially in recent years and understanding the environmental impacts on starch characteristics will contribute to advances for the sector. Herein, the effects of the growing season on the structural, morphological, and physicochemical properties of sweet potato starches were evaluated. Sweet potato trials with two Brazilian cultivars (Canadense and Uruguaiana) were installed in the dry season (planting in March and harvesting in July) and rainy season (planting in October and harvesting in March). Regardless of the cultivar, starches isolated from plants grown in the rainy season have a more ordered structure, with higher gelatinization temperatures, thermal stability, and resistant starch content. Starches from plants grown in the dry season have a higher percentage of small granules with lower crystallinity and lower gelatinization temperatures. These findings can be useful as early knowledge of these changes can help the supply chain to better plan and target suitable markets for naturally modified sweet potato starches.

**Keywords:** *Ipomoea batatas* (L.) Lam; growing season; native starch; physicochemical properties

## 1. Introduction

Sweet potato is an important crop for global food security and has been considered an emerging source of starch. In 2021, the world produced about 88.87 million tons (Mt) of sweet potato roots, with Asia standing out as the largest producer (61.45%), followed by Africa (33.72%) and the Americas (3.78%), with Brazil producing 824,680 tons [1–4].

Starch accounts for 17.5% of the fresh mass and 40–50% of the dry mass in sweet potato roots [5,6]. This polymer is used in a wide range of foods for various purposes, including thickening, gelling, adding stability, and replacing or extending the more expensive ingredients [7].

In the last decade, studies have reported an increase in the use of sweet potato starch in the food industry. The global sweet potato starch market is expected to grow by 3.9% over the next years, from USD 560 million in 2019 to USD 710 million in 2024 [8,9].

The sweet potato growth cycle (4 to 6 months) is characterized by an initial phase (adventitious root growth), an intermediate phase (root tuberization) and a final phase (accumulation of photo assimilates in the tuberous roots). Although sweet potato is a moderately drought tolerant crop, water stress affects plant development by limiting photosynthetic activity, and affecting storage root development, volume, and yield [10,11]. Favorable environmental conditions can lead to an early onset of tuberization, prolonging the period of reserve accumulation in the roots, with increased starch accumulation rate and improved productivity [5]. The process of starch biosynthesis in plants involves isoforms

of several enzymes. In addition to genotypic interference and climate variations from year to year, changes in planting or growing seasons at various locations can have a strong influence on the action of synthesis enzymes and thus on starch functionality [12–14].

Due to the different applicability requirements of starch, chemical modification has been applied. However, currently a greater number of consumers are concerned about their health and have avoided the consumption of products that contain modified starch as an additive on their labels, which has increased the demand for natural starches, which are considered as ingredients [15–18].

The effects of climate change on starch synthesis in plants, structure and functional properties of starches are still poorly explored. In this study, starches isolated from sweet potato plants grown in the dry and rainy season were analyzed for morphological, structural, and physicochemical properties. The results will help broaden the understanding of the impacts of climatic conditions on starch properties, encouraging farmer and industry integration to add value to naturally modified sweet potato starches.

## 2. Materials and Methods

### 2.1. Cultivars, Experimental Area, and Treatments

In this study, the cultivars Canandese and Uruguaina were evaluated. These are the main cultivars planted in the state of São Paulo, Brazil.

Experimental trials were installed at São Manuel Experimental Farm of the São Paulo State University, São Manuel city, SP, Brazil. ($22°44'28''$ S, $48°34'37''$ W, at an altitude of 740 m a.s.l). The region has a Cwa climate (tropical with a dry winter and a hot, rainy summer) according to the Köppen classification system. The soil in the experimental area was classified as a sandy textured dystroferric Red Latosol, corresponding to a dystrophic Typic Hapludox. Prior to the installation of the experiments, soil samples were collected at a depth of 0–20 cm and the chemical characteristics of the soils were determined: pH in $CaCl_2$, 4.8; Soil organic matter, 13 g $dm^{-3}$; $P_{resin}$, 12 mg $dm^{-3}$; H + Al, 26 mmol $dm^{-3}$; K, 2.9 mmol $dm^{-3}$; Ca, 11 mmol $dm^{-3}$; Mg, 4 mmol $dm^{-3}$; ECEC, 43 mmol $dm^{-3}$.

The first planting period was in March and the harvest was in July 2018 (dry season—DS), and the second planting period was in October 2018 and the harvest was in March 2019 (rainy season—RS). Climatic variables were monitored and presented in Figure 1.

At planting, branches of healthy plants with a length of 40 cm containing about eight internodes were selected. The branches were planted at a depth of 10 to 12 cm. The experimental plots consisted of six lines of 5 m in length (0.80 m spacing between lines and 0.3 m between plants). The four central lines were considered as the useful area of the plot, disregarding 0.5 m at the ends of each line. Soil preparation, fertilization for planting and coverage, and phytosanitary management were carried out in accordance with the recommendations for the crop [19] (Figure 2). The plants were harvested at 165 days after planting.

### 2.2. Starch Isolation

The sweet potato roots were washed, peeled, and disintegrated in an industrial stainless-steel blender. The suspension passed through 80-mesh (0.177 mm) and 150-mesh (0.105 mm) aperture sieves to rinse the fibrous residue. The residue retained on the 80-mesh sieve passed again through the same procedure to rinse the residual starch. The recovered starch suspension was mixed into the first suspension and kept in a cold chamber for decantation at 5 °C for 12 h. Afterwards, the recovered starch was siphoned, rinsed in distilled water, recovered by centrifugation, and dried in an oven with air circulation for 12 h at 40 °C [20].

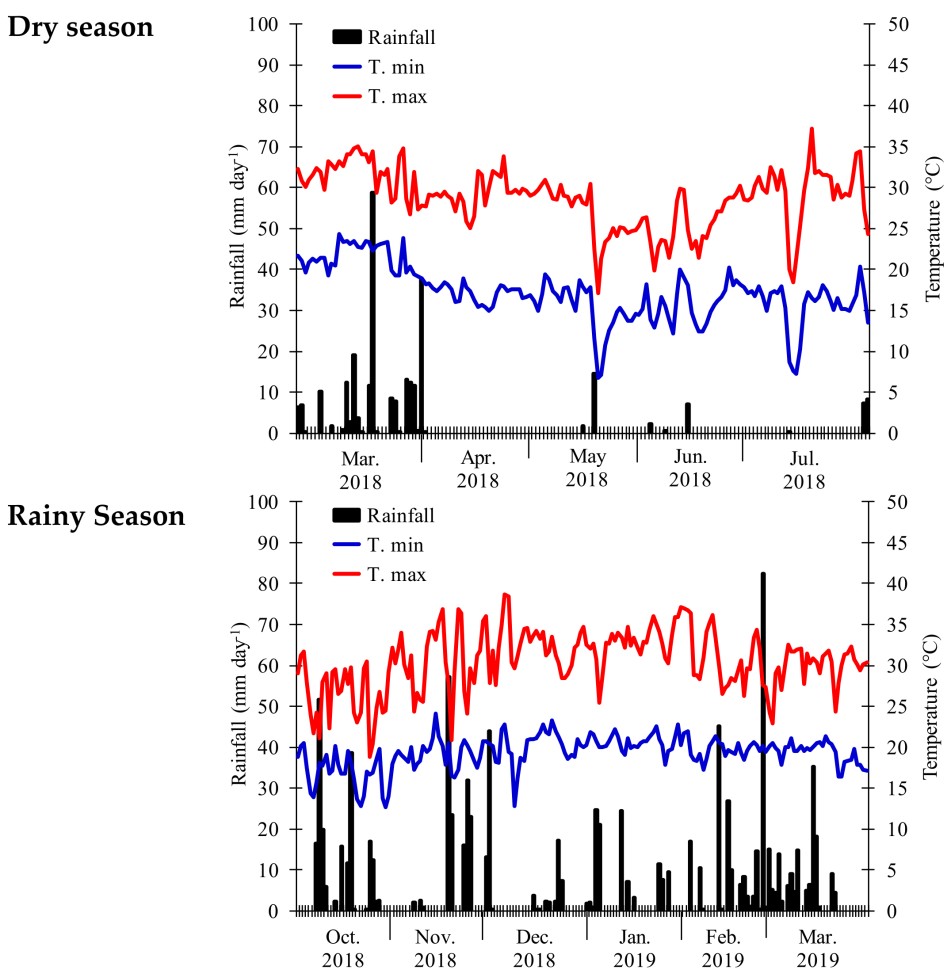

**Figure 1.** Rainfall, maximum and minimum air temperatures recorded in experimental area.

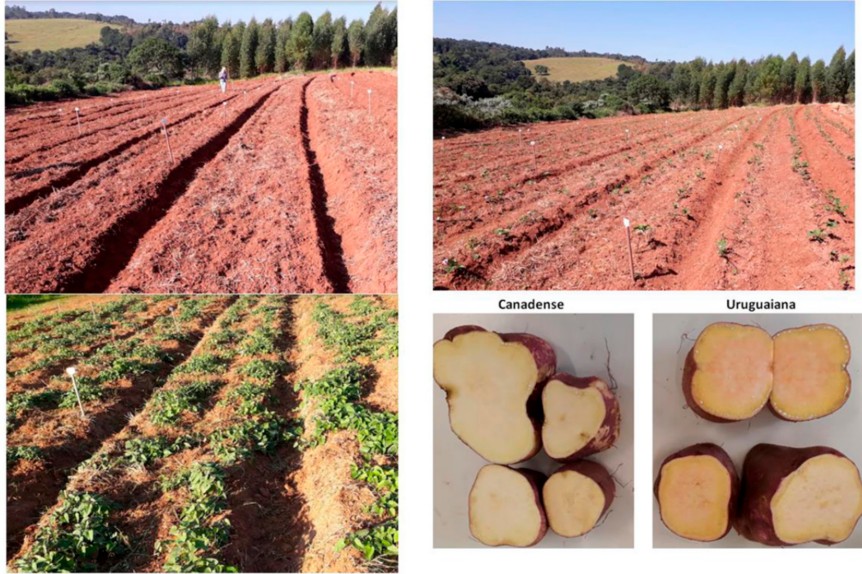

**Figure 2.** Images of the experimental area and roots of sweet potato cultivars.

*2.3. Starch Analysis*

2.3.1. Morphology and Granule Size

The starch granule morphologies were evaluated using scanning electron microscopy (model EVO LS15, Carl Zeiss, Oberkochen, Germany). Starch samples were applied to an aluminum stub with double-sided tape and covered with a thin gold layer (20 nm) in a metallizer for 220 s (Sputter Coater SCD 050- Balzers). The images were obtained using $2000\times$ magnification in high vacuum ($10^{-3}$ Pa) and recorded through the Finepix digital camera and smart SEM software (Carl Zeiss, Oberkochen, Germany).

Starch granule size and size distribution were determined through laser diffraction analysis by using a Helium–Neon laser (Mastersizer 2000, Laser Scattering Spectrometer, Model MAM 5005-Instruments Ltd., Worcestershire, UK). Starch samples were dispersed in distilled water until an obscuration of 5.5% was reached. The refractive indexes of starch samples and solvent were 1.500 and 1.360, respectively. Surface-weighted diameter (D[3,2]), volume-weighted diameter (D[4,3]), median particle size (D[0.5]), and size distribution of the particles were obtained and chosen as granule size through the manufacturer's software (Malvern Application version 5.6, Malvern Instruments Ltd., Worcestershire, UK) [20].

2.3.2. X-ray Diffraction Pattern (XRD) and Relative Crystallinity (RC)

Starch samples were incubated in a desiccator containing saturated $BaCl_2$ solution (25 °C, aw = 0.9) for 10 days to reach humidity equilibrium (90%) and improve the quality of the diffraction diagram. X-ray patterns were examined using the goniometer system Rigaku MiniFlex 600 powder X-ray diffractometer (Cu K$\alpha$ radiation, $\lambda$ = 0.1542 nm) (Rigaku, Tokyo, Japan). The scanning speed was 5 °$\theta$ min$^{-1}$ and the irradiation was performed at 40 kV and 15 mA. The relative crystallinity was calculated based on the relation between the peak and the total area of the diffractogram [21].

2.3.3. Amylose and Resistant Starch

The amylose content of the starch was determined using the method described by Williams et al. [22]. A starch sample (20 mg) was taken, and 10 mL of 0.5 N KOH was added to it. The suspension was thoroughly mixed. The dispersed sample was transferred to a 100 mL volumetric flask and diluted to the mark with distilled water. An aliquot of this solution (10 mL) was pipetted into a 50 mL volumetric flask and 5 mL of 0.1 N HCl was added followed by 0.5 mL of iodine reagent. The volume was diluted to 50 mL and the absorbance was measured at 625 nm.

Resistant starch content was determined according to Goñi et al. [23]. The samples were subjected to: incubation (40 °C, 60 min, pH 1.5) with pepsin (0.1 mL (10 mg/mL), Sigma P-7012) for protein removal; incubation (37 °C, 16 h, pH 6.9) with $\alpha$-amylase (1 mL (40 mg/mL), Sigma A-3176) to hydrolyze digestible starch; residue treatment with 2 M KOH for solubilization of resistant starch; incubation (60 °C, 45 min, pH 4.75) with amyloglucosidase (80 mL (140 U/mL), Sigma A-7255) to hydrolyze the resistant starch solubilized; and the glucose contents in the mixture were measured using glucose oxidase and peroxidase assay kits (GAGO-20, Sigma–Aldrich Company, Saint Louis, MO, USA).

2.3.4. Swelling Power (SP) and Solubility (SS)

The starch samples (0.2 g, wet basis) were placed in tubes and 20 g of distilled water was added based on the initial moisture content. The suspension tubes were immersed in a water bath under constant agitation for 30 min at 95 °C. All tubes were covered with plastic to prevent water loss. Each sample was then centrifuged at $2000\times$ *g* for 15 min; an aliquot (mL) of the supernatant was then collected and left to dry in an oven at 105 °C until constant weight was reached (Ws). The precipitated paste was separated from supernatant and weighed (Wp) [24].

$$SS\ (\%) = [Ws/\ sample\ weight\ (dry\ basis)]\ \times\ 100$$
$$SP\ (g\ g^{-1}) = [Wp\ \times\ 100]\ /\ [sample\ weight\ (dry\ basis)\ \times\ (100\ -\ \%\ S)]$$

### 2.3.5. Pasting and Thermal Properties

The pasting properties of sweet potato starches were analyzed using a Rapid Visco Analyzer (RVA), RVA-4500, Newport Scientific Pty. Ltd., Warriewood, Australia), using Thermocline for Windows, version 3.0. For the analysis, 3 g of each sample were weighed according to their respective moistures, adding approximately 25 g of water to reach a concentration of 10% starch, and were placed in the sample holder of the equipment. For approximately 10 s, the mixture was stirred at 960 rpm (160 rpm during the test). The temperature program used was STD 1. The samples were held at 50 °C for 1 min, followed by heating from 50 °C to 95 °C at a rate of 6 °C minutes$^{-1}$; holding at 95 °C for 5 min, and cooling at 50, at 6 °C min$^{-1}$. The equipment generated viscosity in Rapid Visco Units (RVU), where one unit is equivalent to 12 cP [25].

Thermal properties were evaluated using a differential scanning calorimeter (Perkin Elmer DSC-8500, Norwalk, CT, USA). Starch samples (2.0 mg, dry starch) were mixed with distilled water (6.0 μL) in aluminum pans. The pans were sealed and kept for 2 h at room temperature until balanced; an empty sealed pan was used as reference. The scanning temperature range was 25 °C to 100 °C and the heating rate was 10 °C min$^{-1}$. The equipment was calibrated with indium. The thermal parameters including the temperature of the onset (To), peak (Tp), conclusion (Tc), and the enthalpy change (ΔH) were obtained using the software Pyris 1 (Perkin Elmer, EUA) [20].

### 2.4. Data Analysis

Analysis of variance (ANOVA) was performed with a significance level of 5% and differences between means were determined by Tukey's test, using the Sisvar program (Lavras, MG, Brazil). All measurements were performed in quadruplicate and data are presented as mean and standard deviation.

## 3. Results and Discussion

### 3.1. Morphology and Granule Size

Starch granule morphology is related to amyloplast biochemistry and plant source. For both cultivars grown in dry season (DS), the microscopic analysis showed granules with smooth surfaces, concave and convex polygonal shapes with curved sides with some depression points and a number of small granules (Figures 3 and 4). In the rainy season (RS), the shape of the starch granules was similar to that of the DS, but with a predominance of the rounded shape and a reduced number of small granules (Figure 3). Some authors have already observed similar shapes of sweet potato starch granules [8,26–28].

Starches isolated from plants grown in the rainy season were larger, indicating the interference of water availability on root tuberization and distribution of granule sizes (Table 1). Teerawanichpan et al. [29] compared cassava plants grown in different climates and observed that starch granules were larger in plants grown under higher rainfall than in plants under dry season. In the dry season, there are fewer hours of light; nighttime temperatures are low, in addition to low precipitation, interfering with the rate of photosynthesis and root tuberization. Consequently, there is a higher percentage of granules that are not fully formed.

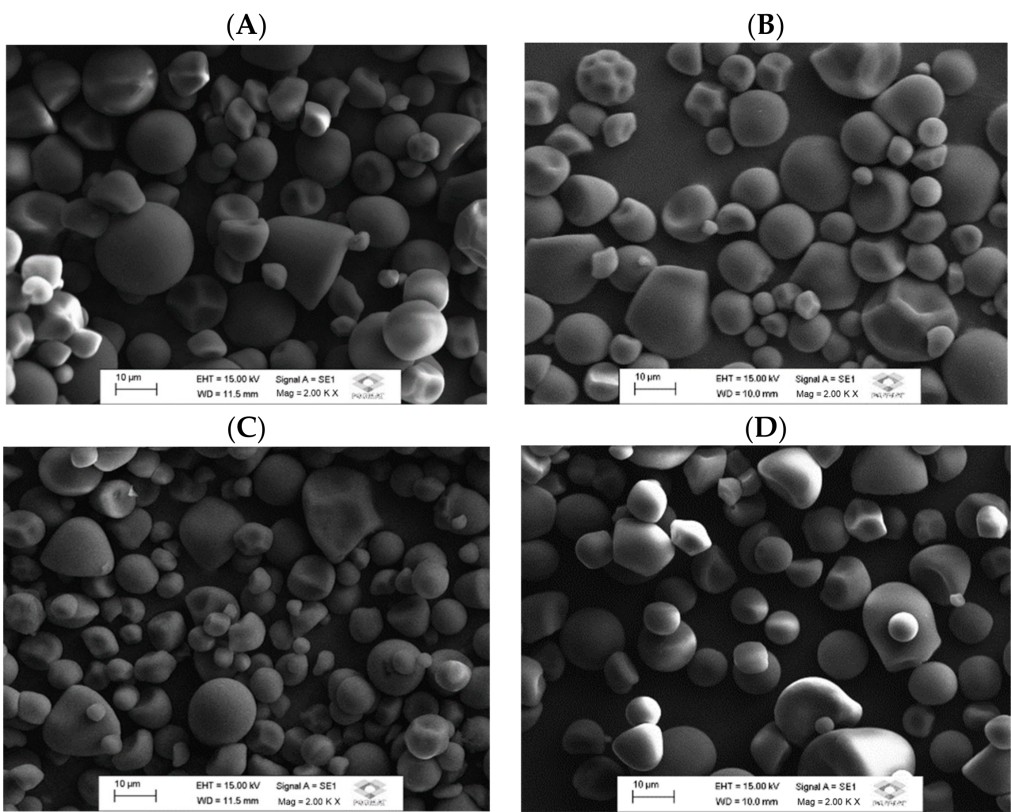

**Figure 3.** Microphotograph of starch granules from sweet potato cultivars. (**A**) Starch from cultivar Canadense (Dry season); (**B**) Starch from cultivar Canadense (Rainy season); (**C**) Starch from cultivar Uruguaiana (Dry season); (**D**) Starch from cultivar Uruguaiana (Rainy season).

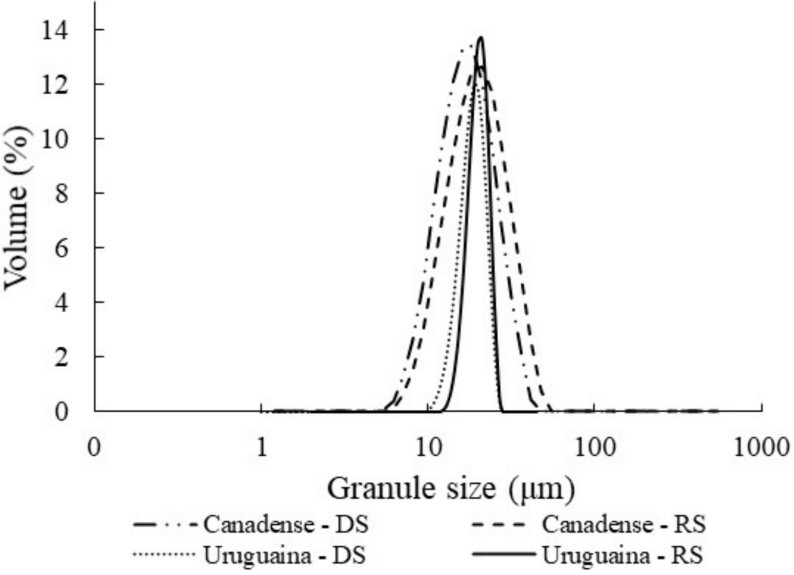

**Figure 4.** Granule size distribution of sweet potato starches. DS = dry season; RS = rainy season.

**Table 1.** Effect of growing season on sweet potato starches in terms of particle size parameters, amylose content, relative crystallinity, resistant starch, swelling power and solubility, and analysis of variance.

| | Canadense | | Uruguaiana | | ANOVA Source of Variation | | |
| --- | --- | --- | --- | --- | --- | --- | --- |
| | Dry Season | Rainy Season | Dry Season | Rainy Season | C | GS | CxGS |
| D[4,3] (μm) | 19.29 ± 0.12 bA | 22.61 ± 0.02 aA | 14.68 ± 0.24 bB | 17.14 ± 0.22 aB | *** | *** | ** |
| D[3,2] (μm) | 16.72 ± 0.14 bA | 19.28 ± 0.02 aA | 12.23 ± 0.11 bB | 14.93 ± 0.19 aB | *** | *** | ns |
| D(0.5) (μm) | 18.11 ± 0.12 bA | 21.13 ± 0.02 aA | 13.49 ± 0.16 bB | 16.12 ± 0.20 aB | *** | *** | * |
| Relative crystalinity (%) | 25.67 ± 0.30 bB | 29.92 ± 0.77 aB | 26.96 ± 0.32 bA | 31.13 ± 0.40 aA | ** | *** | ns |
| Amylose (%) | 25.49 ± 5.6 aB | 25.35 ± 1.1 aB | 26.29 ± 2.0 aA | 25.61 ± 1.3 bA | * | ns | * |
| Resistant Starch (%) | 59.16 ± 9.7 bA | 64.34 ± 6.2 aA | 54.88 ± 9.8 bB | 69.22 ± 2.2 aB | *** | *** | ns |
| Swelling Power (g g$^{-1}$) | 40.09 ± 1.26 aA | 27.18 ± 0.15 bB | 39.07 ± 1.31 aA | 30.56 ± 0.13 bA | * | *** | ** |
| Solubility (%) | 33.38 ± 0.30 aA | 18.48 ± 0.05 bB | 32.15 ± 1.37 aA | 20.39 ± 0.33 bA | ** | *** | ns |

The same lower-case letter in line, in each cultivar, indicates that the results do not differ statistically between growing season; and the same upper-case letter in line, in each season, indicates that the results do not differ statistically between cultivars using the Tukey's HSD test ($p < 0.05$). D[3,2], surface-weight diameter; D[4,3], volume-weighted diameter; D(0.5), median particle size. The means are based on four repetitions. C = cultivar; GS = growing season. ns = Non-significant at the 0.05 probability level; * = Significant at the 0.05 probability level; ** = Significant at the 0.01 probability level; *** = Significant at the 0.001 probability level.

Regardless of the growing season, starches isolated from 'Canadense' had a higher average size, with a wider size distribution. Starches from 'Uruguaiana' had a uniform distribution of granule sizes. These differences may be related to the root system of the cultivar, which interferes with nutrient uptake, storage root formation and tuberization, and adaptation to different climatic conditions, interfering with starch biosynthesis and granule size.

The size of sweet potato starch granules is quite variable. Guo et al. [30] analyzing the granule size distribution of starches extracted from sweet potatoes with white, yellow, and purple pulp observed that the granule sizes ranged from 12.33 to 18.09 μm. Wang et al. [31] after analyzing starches obtained from colored sweet potato varieties observed that of the eight varieties analyzed, three starches showed a bimodal size distribution, with small granules (1–4 μm) and large granules (5–84 μm). The other starches showed unimodal distribution with a granule size ranging from 4.5 to 84 μm. The average size of the starch granules ranged from 16.10 μm to 23.94 μm.

The relationship between the size of granules and the applicability of starches is important. Smaller granules have been valued in edible products such as sauces and dairy desserts, which require a soft mouth feel. They can also be used as fat substitutes due to their similar size to lipid mycelia. Other applications where granule size is important are the production of biodegradable plastic films, paper coatings, and cosmetic products.

*3.2. X-ray Diffraction Pattern and Granule Size*

Starch granules have crystal structures with specific X-ray diffraction patterns, called A, B and C, due to the packing of the double helices of amylopectin. The crystal structure of sweet potato starch is variable and may present patterns of types A, C or C$_A$ [31–33]. Starch isolated from 'Canadense' grown in rainy season showed an A-type diffraction pattern with the most intense peaks at 15, 17, 18, and 23° 2θ. The same cultivar presented a C$_A$-type in dry season with the most intense peaks at 15, 17, and 23° 2θ and a shoulder peak at approximately 18° 2θ, that is, an indicative of great similarity to A-type polymorph. Starch from 'Uruguaiana' had a C$_A$-type diffraction pattern in both seasons (Figure 5).

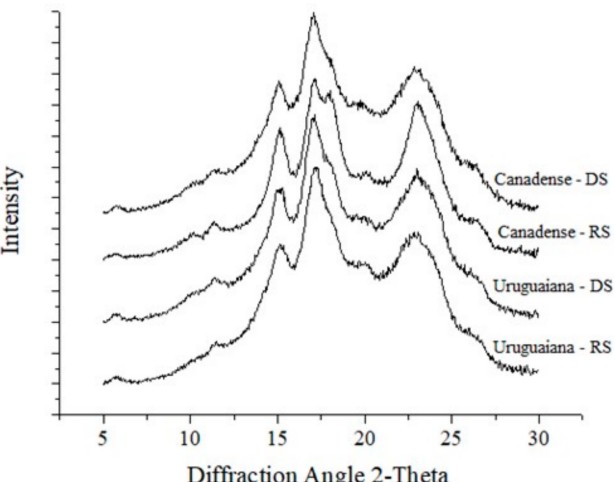

**Figure 5.** X-ray diffraction patterns of sweet potato starches. DS = dry season; RS = rainy season.

Genkina et al. [34] reported the impact of sweet potato crop soil temperature on starch properties. The starches isolated from plants grown at 33 °C had an A-type polymorph, while the isolate from plants grown at 15 °C had a C-type polymorph. These results contribute to the understanding of the pattern variation observed in the starches of the 'Canadense', since in the rainy season daytime and nighttime temperatures are higher than in the dry season, which influences the soil temperature.

Starch is a semi-crystalline material, and the degree and type of crystallinity depends mainly on the structural characteristics of amylopectin. Starches with A-type have a high proportion of short chains in amylopectin. Starches with B and C-type polymorph amylopectin are highly branched, forming long chains linked to amylose molecules. Starches from 'Uruguaiana' showed higher relative crystallinity than those from 'Canadense'. The crystallinities of starches isolated from plants grown in the rainy season were about 4% higher than those of starches from plants grown in the dry season (Table 1). Starches from plants grown in the dry season showed percentages of crystallinity within the ranges reported in other studies [27,28]. The changes resulting from the growing seasons may be related to the actions of starch synthesizing enzymes [28]. Starch crystallinity affects the physical, mechanical, and technological properties of various starchy products, and is therefore important for product development, quality, and process control [25].

*3.3. Amylose and Resistant Starch*

The amylose content and the characteristics of the particles and microstructure of the granules determine whether the starch can be used as a stabilizer, gelling agent or thickener in industries. The amylose content of sweet potato starch, according to several studies ranges from 15.3 to 28.8% [20,26,28,33,35–37].

The amylose content of sweet potato starches varied between cultivars in the dry season, with the highest content observed for starch from 'Uruguaiana' (Table 1). Teerawanichpan et al. [29] observed that the change in amylose content with the growing season was specifically related to the cultivar.

Starches with a high content of amylose are capable of forming inclusion complexes with food ingredients such as essential oils, fatty acids and flavoring molecules, acting as an encapsulant that contributes to an increase in the shelf life of products. In addition, high-amylose starches have interesting nutritional properties, since high amylose is linked to high levels of resistant starch in processed starchy foods [38]. Starches isolated from plants of the same cultivar grown in the rainy season showed higher levels of resistant starch. The same response was observed for starch granule sizes showing that larger granules were more resistant to hydrolysis (r = 0.95, $p < 0.001$). Furthermore, it was possible to verify that starches with higher crystallinity obtained in the rainy season also had a higher resistant starch content, with a positive correlation between the two characteristics

(r = 0.73, *p* < 0.01). Starches isolated from 'Canadense' showed higher levels of resistant starch than those isolated from 'Uruguaiana', which improved its functional properties.

Higher levels of resistant starch can increase the commercial value of natural sweet potato starch. Research indicates that the resistant starch market was valued at USD 10.5 billion in 2022 and is expected to reach USD 19.9 billion by 2032 (CAGR of 6.6%). This market has been driven by growing awareness of the health benefits of resistant starch and increasing use of resistant starch in bakery products, confectionery products, dairy products, breakfast cereals, beverages, among others [39].

### 3.4. Swelling Power (SP) and Solubility (SS)

In gelatinization, the starch structure breaks down, leading to the weakening of hydrogen bonds and the interaction of water molecules with the hydroxyl groups of amylose and amylopectin, causing swelling and partial solubilization of the starch.

Starches isolated from plants grown in the dry season had higher SP and SS than those isolated from plants grown in the rainy season. Starch from the 'Uruguaiana' had higher SP and SS in the rainy season (Table 1). The values observed in this study for starches isolated in the dry season were close to those observed in other studies conducted in Brazil with sweet potato starches [27,40], and those obtained in the rainy season were similar to those observed in studies conducted in China [30,37].

The effects of the dry season on SP and SS are consistent with the smallest granule size (r = −0.99, *p* = 0.01), higher amylose (r = 0.79, *p* < 0.05), and lower resistant starch (r = −0.81, *p* < 0.05) observed in starches isolated from plants grown in these climatic conditions. The relationship between these physicochemical parameters, and the water absorption capacity and solubilization of starches has already been reported by Guo et al. [30].

### 3.5. Pasting and Thermal Properties

Data analysis of pasting properties of sweet potato starches showed a greater effect of the growing seasons on the viscosity parameters (Table 2, Figure 6). Starches from 'Canadense' have higher viscosity peaks and breakdown and starches from 'Uruguaina' have higher retrogradation tendencies.

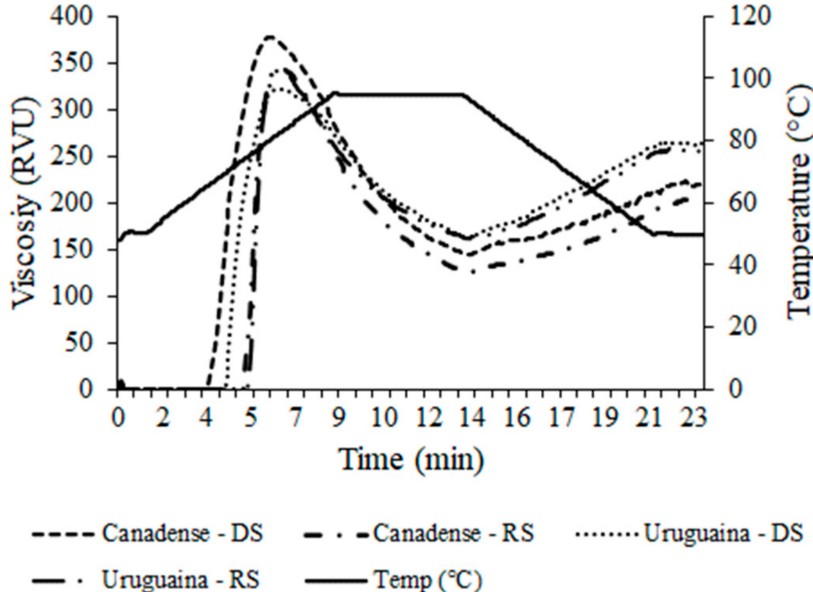

**Figure 6.** Rapid Visco Analyzer (RVA) profiles of sweet potato starches. DS = dry season; RS = rainy season.

**Table 2.** Effect of growing season on sweet potato starches in terms of paste properties, and analysis of variance.

| | Canadense | | Uruguaiana | | ANOVA Source of Variation | | |
| --- | --- | --- | --- | --- | --- | --- | --- |
| | Dry Season | Rainy Season | Dry Season | Rainy Season | C | GS | CxGS |
| Peak Viscosity (RVU) | 382.08 ± 13.92 aA | 315.42 ± 0.34 bB | 338.47 ± 5.10 aB | 324.25 ± 2.22 bA | ** | ** | *** |
| Breakdown (RVU) | 234.22 ± 10.24 aA | 193.63 ± 1.54 bA | 177.55 ± 3.30 aB | 156.97 ± 5.17 bB | *** | ** | *** |
| Final Viscosity (RVU) | 222.39 ± 9.94 aB | 198.71 ± 2.13 bB | 251.08 ± 4.63 bA | 271.92 ± 8.81 aA | *** | ** | ns |
| Setback (RVU) | 74.53 ± 1.08 bB | 76.92 ± 0.25 aB | 90.17 ± 1.28 bA | 107.75 ± 0.82 aA | *** | * | ** |
| Pasting Temperature (°C) | 66.23 ± 1.18 bB | 75.63 ± 0.03 aA | 69.57 ± 0.45 bA | 75.38 ± 0.42 aA | * | *** | ** |

The same lower-case letter in line, in each cultivar, indicates that the results do not differ statistically between growing seasons; and the same upper-case letter in line, in each season, indicates that the results do not differ statistically between cultivars using the Tukey's HSD test ($p < 0.05$). Rapid Visco Unit (1 RVU = 12 cP). The means are based on four repetitions. C = cultivar; GS = growing season. ns = Non-significant at the 0.05 probability level; * = Significant at the 0.05 probability level; ** = Significant at the 0.01 probability level; *** = Significant at the 0.001 probability level.

The resistance to mechanical action during the period of constant temperature maintenance in the RVA analysis indicates that the starch has strong intermolecular bonds. Granules with low swelling power are more resistant to prolonged heating and/or mechanical agitation, therefore, less susceptible to granule rupture, which is related to viscosity stability. Starches isolated from plants grown in the rainy season had higher crystallinity, lower swelling power and lower viscosity breakdown. This type of starch is preferred as a thickener in foods that require a long heat treatment time under agitation, for example, in processes that involve treatment with high temperatures and pressure, such as autoclaving, and manufacture of soups and canned products [41].

All starches showed an increase in viscosity on cooling, with the highest setback values observed for starches isolated from the 'Uruguaiana' in the two growing seasons of the plants. Regardless of the cultivar, the setback was positively correlated with the amylose content in the two growing seasons (Dry season, r = 0.73, $p < 0.05$; Rainy season, r = 0.88, $p < 0.05$). Setback viscosity is an indirect measure of starch retrogradation tendency. Retrogradation is the process of crystallization of starch chains, particularly amylose molecules, which occurs after the gelatinized starch paste has cooled, forming a cohesive three-dimensional network. The higher setback indicates lower stability of the starch paste in cold and this parameter allows the estimation of the stability of the starch gel during storage at low temperatures, considering that starches with lower tendencies to retrogradation are more desirable by the food industry.

The gelatinization process depends mainly on the dissociation of the helical structure within the starch chains, and the energy required to dissociate this structure varies with different starch sources. The results observed in the analysis of the thermal properties showed that both cultivars had higher gelatinization temperatures and enthalpy range in the rainy season (Table 3, Figure 7). Starches isolated from 'Uruguaiana' had higher gelatinization temperatures and enthalpy range in the dry season, differing in all parameters from those isolated from 'Canadense'. In the rainy season, the cultivars differed in the initial and peak temperatures.

**Table 3.** Effect of growing seasons on sweet potato starches in terms of thermal properties, and analysis of variance.

| | Canadense | | Uruguaiana | | ANOVA Source of Variation | | |
| --- | --- | --- | --- | --- | --- | --- | --- |
| | Dry Season | Rainy Season | Dry Season | Rainy Season | C | GS | CxGS |
| $T_{onset}$ (°C) | 57.47 ± 0.19 bB | 70.32 ± 0.20 aB | 58.10 ± 0.17 bA | 71.72 ± 0.05 aA | *** | *** | ** |
| $T_{peak}$ (°C) | 62.07 ± 0.07 bB | 74.69 ± 0.20 aB | 63.30 ± 0.19 bA | 75.31 ± 0.14 aA | *** | *** | ** |
| $T_{conclusion}$ (°C) | 68.23 ± 0.09 bB | 79.28 ± 0.10 aA | 70.50 ± 0.41 bA | 79.34 ± 0.23 aA | *** | *** | *** |
| $T_{conclusion}$-$T_{onset}$ (°C) | 10.77 ± 0.27 aB | 8.96 ± 0.15 bA | 12.40 ± 0.41 aA | 7.62 ± 0.19 bB | *** | *** | *** |
| $\Delta H$ (J g$^{-1}$) | 12.51 ± 0.29 bB | 14.84 ± 0.19 aA | 13.94 ± 0.26 bA | 14.85 ± 0.30 aA | * | *** | ** |

T, temperature; $T_{conclusion}$-$T_{onset}$, temperature range ($\Delta T$); $\Delta H$, enthalpy change. C = cultivar; GS = growing season. The means are based on four repetitions. ns = Non-significant at the 0.05 probability level; * = Significant at the 0.05 probability level; ** = Significant at the 0.01 probability level; *** = Significant at the 0.001 probability level.

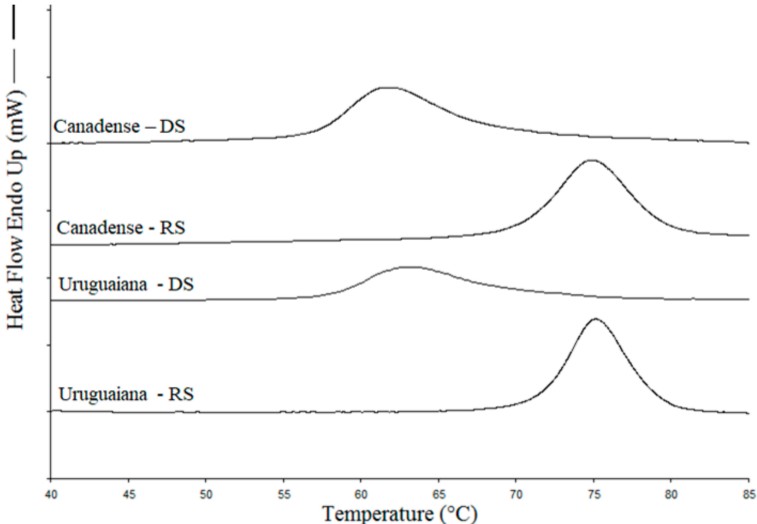

**Figure 7.** Differential scanning calorimetry (DSC) curves of sweet potato starches. DS = dry season; RS = rainy season.

Tsakama et al. [42] analyzed the pasting properties of eleven sweet potato genotypes and observed that the pasting temperature ranged from 73.4 to 75.88 °C, the peak viscosity ranged from 1947 to 2596 cP, the hot viscosity ranged from 1496 to 2049 cP, the breakdown ranged from 221 to 889 cP, the cold viscosity ranged from 2304 to 2762 and the setback ranged from 1.51 to 1.71 cP.

Gelatinization temperature provides a measure of granule crystallinity [43], and in this study, differences of approximately 12 °C in gelatinization temperatures were observed between the growing seasons for the two cultivars, indicating a possible presence of a larger area of crystallinity in starches isolated from rainy season plants in relation to the dry season. The higher gelatinization temperatures and enthalpy range can be explained by the higher relative crystallinity of the starch granules, which provides a higher structural stability [44].

Campanha and Franco [45] reported gelatinization temperatures of sweet potato starch ranging from 62.9 to 77.9 °C, with the peak at 70.6 °C, and an enthalpy range of 12.9 J/g. The gelatinization temperatures of sweet potato starch were higher than those observed for cassava (59.0 to 71.2 °C) and potato (61.9 to 69.9 °C) starches, suggesting stronger crystalline structures and a higher molecular order of sweet potato starch.

The onset, peak, and conclusion temperatures of gelatinization observed in this study were similar to those reported in other studies [8,30,41]. The enthalpy of gelatinization ($\Delta H$) provides a general measure of crystallinity and is an indicator of the loss of molecular

order within the granule during gelatinization. In both growing seasons, the enthalpy of gelatinization of sweet potato starches was positively correlated with the crystallinity of the starches (DS, r = 0.92, *p* < 0.001; RS, r = 0.88, *p* < 0.01). Higher ΔH as observed for starches isolated from the 'Uruguaiana' and those from plants grown in the rainy season suggests a greater degree of organization or greater stability of the crystals.

## 4. Conclusions

Different plant-growing seasons led to the production of naturally modified sweet potato starches, regardless of cultivar. Starches isolated from plants grown in the rainy season have a more ordered structure with higher resistance to thermal processes. Starches isolated from plants grown in the dry season showed lower resistance to heat and agitation, tendency to retrogradation, paste temperature and enthalpy of gelatinization. In addition, dry season starches had a lower content of resistant starch. These differences point to different potential uses of naturally modified starch as an ingredient for food products. With the global sweet potato starch market growing, our findings are also important for defining industrial planning strategies.

**Author Contributions:** Conceptualization, M.L., A.M.F., T.P.R.d.S. and S.L.; methodology, validation, formal analysis, and investigation, T.P.R.d.S., L.A.d.O. and J.G.d.S.N.; data curation, writing—original draft preparation, writing—review and editing, T.P.R.d.S., M.L. and S.L.; supervision, project administration, funding acquisition, M.L. All authors have read and agreed to the published version of the manuscript.

**Funding:** This work was partially supported by the Brazilian National Council for Scientific and Technological Development (CNPq), grant numbers 302611/2021-5 and 302848/2021-5 and Coordination for the Improvement of Higher Education Personnel (CAPES—PNPD scholarship).

**Data Availability Statement:** Data are contained within the article.

**Conflicts of Interest:** The authors declare no conflict of interest.

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
