# Peer review of "Seasonal Variations in the Starch Properties of Sweet Potato Cultivars"

_horticulturae, doi:10.3390/horticulturae9030303_

Round 1

Reviewer 1 Report

The authors presented the article describing the effects of two growing seasons (dry and rainy) on structural, morphological, and physicochemical properties of sweet potato starches isolated from two Brazilian cultivars (Canadense and Uruguaiana). The results presented in this very interesting and well written work can be useful for commercial producers of this valuable worldwide used vegetable.

Introduction part is correctly written with sufficient literature data. The whole experimental methods used in this work are nicely described with all sufficient details. The figures in the manuscript are illustrative and with good quality, which is often very demanding job. The obtained results, discussed sufficient and in details, are followed by summary conclusion.

Author Response

The authors thank you for accepting to review the manuscript and the comments made in order to improve the quality of the work. The text was revised according to the opinion of four reviewers, and the authors revised the text for English language and style.

Reviewer 2 Report

The communication title “Seasonal variations on the starch properties of sweet potato cultivars” is conducted well, and has scientific worth; it needs minor revisions.

Comments for authors:

1-      2.3.3: Amylose and resistant starch protocols/methods are missing.

2-      Line 252, 316, 359, etc., in the figure title, please write the full name of abbreviations. (DRX, RVA and DSC etc.).

3-      Where is ANOVA table?

4-      I suggest author to perform a correlation analysis among the different starch parameters with dry and rainy seasons.

5-      Line 489: Delete.

Author Response

The authors thank you for accepting to review the manuscript and the comments made in order to improve the quality of the work. The descriptions of the methodologies used in the determination of amylose and resistant starch contents were included in the text. Figure titles were revised and ANOVA and Pearson's correlation information were inserted in the tables and in the text.

Reviewer 3 Report

The manuscript entitled “Seasonal variations on the starch properties of sweet potato cultivars” by dos Santos et al. presented a well-conducted study. The manuscript showed the growing seasons/environmental conditions had impact on the structural, physicochemical and functional properties of sweet potato starch. The findings have implications for sweet potato production and starch applications. The data were presented in the logic order and easy to follow. However, I have a few suggestions that might be useful and can improve the manuscript.

1.      In “Results and Discussion”, multiple small paragraph could be re-organized into bigger paragraph.

2.      Figure 6 & 7, provide full name for RS and DS in figure legend.

3.      Careful proofreading is necessary to correct a few grammar errors and typos.

Author Response

The results and discussion were revised in accordance with the suggestion to reduce the number of small paragraphs. The legends of Figures 6 and 7 have been corrected. The text has been revised.

Reviewer 4 Report

Manuscript ID: horticulturae-2204954

Type: Article

Title: Seasonal variations on the starch properties of sweet potato cultivars

Authors:  Thais Paes Rodrigues dos Santos , Magali Leonel * , Luciana Alves de Oliveira , Adalton Mazetti Fernandes , Sarita Leonel , Jason Geter da Silva Nunes

Review of the manuscript

Comments

The manuscript presented for revision is very interesting. This work concerns a important area of science and is of great practical importance. The two Brazilian cultivars (Canadense and Uruguaiana) of sweet potatoes, studied by the authors of this manuscript, are an important source of starch, widely used in many industries. The climatic changes, observed in recent years and the related changes in vegetation conditions affect, as demonstrated in the reviewed manuscript, the structural, morphological, and physicochemical properties of sweet potato starch. The results of these studies can help to better use the raw material (sweet potatoes) for various modifications and applications.

The obtained results are discussed with the works published in recent years. This work is well organized and scientifically sound. 

To sum up, my recommendation is - accept in present form

Author Response

The authors thank you for accepting to review the manuscript and the comments made in order to improve the quality of the work. The text was revised according to the opinion of four reviewers.